# Patterns of Psychiatric Comorbidity Among Drug Users: A Prospective Observational Study in a Romanian Psychiatric Hospital

**DOI:** 10.3390/healthcare13192543

**Published:** 2025-10-09

**Authors:** Andreea Atena Zaha, Antonia Lucia Comșa, Dana Carmen Zaha, Cosmin Mihai Vesa

**Affiliations:** 1Doctoral School of Biomedical Sciences, University of Oradea, 410087 Oradea, Romania; zaha.andreeaatena@student.uoradea.ro; 2County Emergency Hospital, 400347 Cluj-Napoca, Romania; antonialuciacomsa@gmail.com; 3Department of Preclinical Disciplines, Faculty of Medicine and Pharmacy, University of Oradea, 410068 Oradea, Romania; cosmin.vesa@csud.uoradea.ro

**Keywords:** psychoactive substances, dual diagnosis, psychiatric comorbidity

## Abstract

Background: A large number of substance use disorders are increasingly associated with complex clinical presentations and unknown mental and medical risks, presenting a growing challenge for mental health worldwide. Research exploring the interplay between substance use and psychiatric disorders remains limited in Eastern Europe. Objectives: We investigated the demographic and clinical features of 203 patients admitted to a major Romanian psychiatric hospital, aiming to clarify the patterns of dual diagnosis and symptomatology within this vulnerable population. Results: Cannabis, novel psychoactive substances and unknown substances were the most commonly used drugs. Psychiatric comorbidity was rather the rule than the exception within our group. Cluster analysis revealed three distinct symptom profiles: manic/psychotic, negative affective and disorganized. While individual drug type did not independently predict symptom severity or readmission risk, a significant interaction effect between drug use and psychiatric comorbidity influenced symptom cluster membership. Conclusions: These findings highlight the complexity and heterogeneity of dual diagnoses and underline the importance of an integrated, multidisciplinary approach in addiction medicine.

## 1. Introduction

Substance abuse has been substantially on the rise during the past three decades, worldwide and in Europe, resulting in a wide range of social, medical and financial issues. Due to the growing diversity of novel substances, the trends of consumption have been rapidly changing, which is associated with significant clinical unpredictability and raises important questions for mental health professionals [1]. Patients with dual diagnosis are a very vulnerable category, requiring special assessments of prognosis and treatment modalities. While we know that substance use disorders raise the risk for psychiatric disorders, the complex interplay between them, as well as the actual risks, remains partially a mystery due to two important factors: gaps in knowledge over classic illicit substances, on the one hand, and the emergence of novel psychoactive substances (NPS), on the other hand.

Amphetamine use disorder has also been linked to psychosis (aOR = 2.4, 95%CI 1.6–3.5), as well as to suicidality (aOR = 1.5, 95%CI 1.3–1.8) and violence (OR = 6.2, 95%CI 3.1–12.3) [2]. Chronic methamphetamine use has been identified as a risk factor for cognitive and self-regulatory impairments, namely in the inhibitory control domain [3].

Psychedelics have also recently been studied closely, after a decades-long lack of research due to restrictions. They have been shown to rapidly and lastingly increase neuroplasticity, offering potential beneficial effects in treating addictions, anxiety, depression and traumas [4]. Concerns regarding risks of developing schizophrenia remain, even though a large meta-analysis has identified a 0.002% rate of psychedelic-induced psychosis, out of which 13.1% later developed schizophrenia [5].

The second factor, the emergence of NPS, is an important public health issue as new substances come into the market at an alarmingly fast pace. In 2021, 1124 new substances were reported [6]. Adverse health effects include a wide range of fatalities, suicide and infectious diseases; however, these are still poorly researched due to the overwhelming range of substances [7]. Synthetic cathinones are the most consumed types of NPS, together with cannabinoids, and they have been associated with various cardiac and neurological effects, including agitation, severe psychosis, hyperthermia, encephalopathy, coma and convulsions, sympathomimetic and hallucinogenic toxidromes, excited delirium and serotonin syndrome [8]. The long-term effects on general and psychiatric health remain unknown. Our hypothesis is that psychopathology and prognosis are worse for NPS users compared to non-NPS users.

To provide a coherent framework for understanding the interplay between substance use and psychiatric symptomatology, this study draws upon the self-medication hypothesis and neurotoxicity theory within the broader context of the vulnerability/stress model. The vulnerability/stress model posits that the interaction between biological vulnerability and environmental factors causes psychiatric symptoms. On the other hand, individuals with psychiatric vulnerability may cope with distressing symptoms using substances, a theory known as the self-medication hypothesis [9]. Neurobiological changes due to chronic drug use (especially of poorly known and regulated substances such as NPS) may further exacerbate psychiatric symptoms through neurotoxicity and neurotransmitter dysregulation. The interaction between all these factors forms the theoretical basis for our focus on dual diagnosis, symptom clustering and drug/comorbidity interactions.

Regional differences in drug consumption patterns give a unique risk profile for (sub)populations. According to the EUDA Romania had the highest rate of novel psychoactive substances (NPS) use in the European Union, with a prevalence of 5.1%. By comparison, in Latvia and Norway, this prevalence was 0.1% [10]. While this study is not a nationwide epidemiological survey, we hope to find clues about the possible explanations of this high rate.

To our knowledge, there have not been any published in-depth studies regarding the overlap between substance abuse and psychiatric disorders in any psychiatric hospital in Romania. A comparison between patterns of use between Romanian and Hungarian patients has been published in 2021, which concluded that Romanian drug users were younger and had a comorbid personality disorder more frequently than Hungarian users, who were more frequently diagnosed with comorbid affective disorders and alcohol use disorders [11].

The current study investigated the demographic and clinically relevant patterns of drug consumption and prognosis of patients admitted to a major Romanian psychiatric hospital, aiming to shed some light on the patterns of dual diagnosis and symptomatology within this vulnerable population. We particularly sought to answer questions regarding clinical differences between genders, between individuals with and without dual diagnosis, and between NPS users and non-users.

## 2. Materials and Methods

### 2.1. Study Design and Data Collection

In order to emphasize the complete picture in a clinical setting, we designed a prospective study for one year, between 1 January 2024 and 31 December 2024. This took place in Cluj-Napoca, Romania, at the Emergency County Hospital, Department of Psychiatry 3. This study was designed as an analytical observational study, where clustering and predictive modelling were used.

The sample included all adult patients admitted with ICD-10 substance use diagnoses in 2024; records with missing data were excluded. Data was collected from unstructured clinical interviews and the internal hospital database, by two of the researchers. The included parameters were demographic characteristics, details about admission (voluntary or involuntary, duration of hospital stay), psychiatric symptoms (presence of specific symptoms, as well as intensity, quantified by the Brief Psychiatric Rating Scale), psychiatric diagnosis, type of drug(s) consumed, legal and social issues, treatment and paraclinical values. Readmission was defined as any psychiatric hospitalization, not only substance-related. When distinguishing between “primary” and “secondary” drugs, the used criterion was current frequency of use.

ICD-10 criteria were used in order to determine the psychiatric diagnoses. The diagnostic process was conducted in a team including psychiatry residents, nurses and psychiatry specialists.

### 2.2. Data Analysis

Psychiatric symptoms were first binary assessed (presence or absence of major groups and subgroups of symptoms) using medical records, then quantified through the BPRS scale. This clinician-rated tool evaluates 18 symptom domains including anxiety, depression, hallucinations and disorganized thinking. Each item is scored from 1 (not present) to 7 (extremely severe). While formal Romanian-language validation studies are lacking, the BPRS has demonstrated excellent psychometric properties (Cronbach’s α = 0.80–0.85; ICC = 0.75–0.90) in a recent study, which included some Romanian participants as well [12]. The scale is widespread used; however, further research could test more in-depth its psychometric properties on specific populations.

Data analysis was performed in SPSS IBM 29.0 and RStudio 2024.12.0+467. Qualitative data were first analyzed with simple frequency functions, followed by creating contingency tables and computing Chi2 and Fisher’s exact test scores. Principal component analysis followed by k-means clustering has been used for investigating patterns of symptoms. A multinomial regression model has been applied to investigate the predictors of these patterns.

No formal sample size calculation was performed, as this study is primarily exploratory and descriptive. The sample size was determined by feasibility and the availability of participants within this hard-to-reach population.

### 2.3. Ethical Approval

This study was conducted under the approval of the Ethics Committee number 57030/6 December 2023.

## 3. Results

### 3.1. Population Descriptives

A total number of 203 patients admitted throughout 2024 were included in this study. Upon admission and throughout it, most of them were cooperative (154, 75.9%). The maximum number of readmissions in the same month per patient was 2. There were 32 patients readmitted at some time during the year, excluding the same month, with a minimum of one and maximum of eight per patient. The average number of readmissions per year was 1.69 ± 1.44. Ages ranged from 17 to 84 years old, with a mean of 31.79 ± 11.94. Most patients are male, unemployed and live in the city, where access to medical services is easy. Over 80% of patients are single (Table 1).

### 3.2. Clinical Features

A summary of the clinical findings can be seen in Table 2. Orientation in time and place was present in 180 patients (147 men, 33 women). Most of them had insight into their illness (56.2%; 114 in total, out of which 86 were men and 28 women), meaning that they recognized the negative mental health outcomes of substance abuse and/or the importance of treatment. This was, however, not described in depth, for example, using a scale. Our patients displayed anxiety, emotional lability and persecution delusions, especially the men (*p* < 0.05). Past and actual depressive episodes were both significantly more frequent for men. The same finding was observed on the frequencies of actual light and moderate depressive episodes. On the other hand, women had significantly higher rates of depressive delusions. Regarding aggression, it also seems to predominate in men.

### 3.3. Profile of Drug Consumption

The specific types of consumed drugs can be seen in Table 3. The mean symptom severity was calculated for NPS and non-NPS users, resulting in the following values: −0.309 and 0.12. A T-test for independent groups showed a t-statistic of −1.422, with a *p*-value of 0.15. The most used primary drugs were cannabis, methamphetamine, benzodiazepines and unspecified.

### 3.4. Psychiatric Diagnoses

The most common primary diagnoses can be seen in Table 4. As primary diagnoses, there was also one of each of the following: harmful use of opioids, cannabis withdrawal syndrome, psychosis induced by cocaine, harmful use of cocaine, other stimulants (including caffeine withdrawal syndrome) and psychosis induced by solvents.

### 3.5. Interplay Between Symptoms and Drug Consumption

Principal component analysis identified 24 distinct components in our dataset, out of which 6 were used for k-means clustering. The scree plot showed three components under and three above the elbow. The three components above the elbow explained most of the variability, which is why these were retained (Figure 1). This yielded three clusters of symptoms (together with the corresponding mean z-scores):Cluster 1 (ManPsy)—characterized by especially high scores of elevated mood (0.131), grandiosity (0.12), excitement (−0.29), distractibility (−0.23) and motor hyperactivity (−0.21). Hostility, disorganized behavior, lack of cooperation and delusions were also present to a smaller extent.Cluster 2 (NegAffect)—the highest scores for depression (−0.18), somatic concern (0.06), guilt, emotional withdrawal (−0.41), motor inhibition (0.1) and suicidality (−0.16) across all the clusters. However, these values were relatively low, suggesting a lower symptom intensity. This group had the lowest scores for psychotic symptoms and conceptual disorganization.Cluster 3 (Disorg)—characterized by the highest scores of hallucinations (0.22), delusions (0.25), disorganized behavior (0.37), formal thought disorder (0.47), suspiciousness, mannerisms (0.24) and self-neglect scores (0.33). In contrast to the first cluster, manic symptom scores were very low.

The between-cluster/total sum of squares ratio was 86.1%, indicating that 86.1% of the total variance in the data can be explained by the clustering. Therefore, the clusters were well defined (Figure 2).

A random forest model was built to determine the predictive power of the main drug consumed and of the main comorbid psychiatric diagnosis on cluster membership, hospital readmission within the month and hospital readmission within the entire year. This model yielded a 68.3% accuracy, which suggested that the predictors can partially account for clinical features, but other factors likely play a part as well. A summary of this model can be seen in Table 5.

ANOVA was then applied to test whether comorbidity and drug type had significant influences on symptoms and readmission. First, each predictor was tested separately, then interaction effects were used to test whether a comorbidity and drug type together would significantly influence the outcomes. The main findings can be seen in Table 6.

Tukey’s HSD post hoc test showed that no specific drug pairs showed significantly different effects on symptoms or readmission. Simple effect analysis showed a significant drug effect within the cannabis dependence group (F = 6.81; *p* = 0.026), indicating that the type of drug consumed had an influence on the different symptom profiles within this group. No other subgroup comparisons reached statistical significance.

The distribution of ICD-10 diagnoses across the clusters is shown in Figure 3. Standardized residuals suggest that cannabis use disorder was strongly overrepresented in the ManPsy cluster (2.38), while opioid use disorder was overrepresented in the NegAffect cluster (1.12) and underrepresented in the Disorg cluster (−1.35). Alcohol and cocaine use disorders showed no significant deviations across clusters.

## 4. Discussion

Our results highlight a mix of predictable and less predictable findings.

A surprising finding was the occupational distribution. University is associated with increased risk for substance use, especially in academic cities, such as Cluj-Napoca [13]. In our study, most of the included patients were either unemployed or (legally) employed. Interpreting this finding in the larger context, we can assume that drug use in our population occurred later or that drug use affected their lives to such an extent that most of them faced unemployment. The reverse can also be true, so it is difficult to establish which one came first—the unemployment or the drug use.

Similar to other studies on drug use, most of our subjects were single. Loneliness and emotional neglect are two factors that have been consistently linked with drug abuse, suicidality and psychiatric illness [14]. While that may be a risk factor for developing an addiction, simple observation also quite often shows difficulties when it comes to interpersonal skills. Although unfortunately not formally analyzed, there is a high rate of systemic issues presented by our subjects.

Age and gender were not associated with readmission and symptom severity. Agitation, irritability and sleep disturbances were the most frequent symptoms at admission. Emotional lability, impulsiveness and formal thought disorders were present in more than half of the patients. Anxiety, aggression, emotional lability, depression and withdrawal rates were significantly higher for men than for women. In our cohort, aggression was predominantly verbal, though physical aggression was also more frequent in men. While no single substance type showed a statistically significant association with aggression, stimulant and cannabis users displayed numerically higher rates, in line with existing research linking these substances to disinhibition and irritability. Depressive delusions were significantly more frequently found in women, supporting Carter’s results, which indicated that women experienced more severe depressive symptoms than men [15]. Anxiety was a predictor of substance abuse in women in other studies and a common finding together with depressive disorders, contrary to our results [16,17]. In the latter study, men had higher rates of externalizing disorders such as antisocial personality disorder and conduct disorder, which is similar to our finding that aggression was more common in men. The high rate of anxiety and depression in men could be explained by the gender distribution in our sample (predominantly men, relatively few women) or perhaps by cultural particularities.

The induced psychosis rate was high, affecting nearly a third of the group, while delusions were present in 40.9% of patients. Persecutory delusions were the most common (25.6%), followed by interpretation (19.7%) and following delusions (16.3%), indicating a clear predominance of paranoid symptomatology. Spiritual and grandiose delusions occurred at similar frequencies (8.4% and 9.4%), whereas depressive, poisoning and jealousy delusions were rare. These strikingly elevated rates of psychotic features exceed those typically reported in general psychiatric populations and are consistent with the impact of cannabis and stimulant use. Beyond complicating differential diagnosis—particularly the distinction between substance-induced and primary psychotic disorders—such symptoms likely contribute to relapse risk and repeated hospitalizations. Notably, 32 patients were readmitted during the study year, some up to eight times. Although our analyses did not directly test the relationship between psychotic symptoms and readmission, the combination of high psychosis and delusion rates with a subgroup experiencing multiple readmissions suggests that these symptoms may contribute to relapse risk. This warrants further study in larger cohorts.

Based on the Brief Psychiatric Rating Scale results, three clusters of symptoms were found—one with manic and psychotic features, one with negative mood and the last with disorganized features. Cannabis use disorders showed strong associations with the ManPsy cluster. According to a systematic review, cannabis use is associated with an approximately threefold (OR: 2.97; 95%CI: 1.80–4.90) increased risk for the new onset of manic symptoms [18]. Opioid use disorder and multiple substance use disorder were associated with the NegAffect cluster. The latter was also the most common diagnosis. The clustering solution showed strong validity, with a high between-cluster variance ratio indicating well-separated groups. The three profiles mirror broader transdiagnostic dimensions, suggesting that substance-related presentations align with recognizable psychopathological structures rather than substance-specific syndromes. While this offers potential clinical utility in anticipating care needs, the relatively small sample size limits the generalizability of these clusters and warrants replication in larger cohorts.

The high frequency of double diagnosis (83.25%) was impressive, underlining the stance that psychiatric comorbidity is rather the rule than the exception. A similar result was found by Davis et al., where 66% of the cohort reported a diagnosis of any mental illness [19]. A high prevalence of severe mental illness in patients presenting to drug treatment facilities has been found by Havassy et al. [20]. Comorbid psychiatric diagnoses played a more significant role in the clinical features and prognosis in comparison to the specific psychoactive substances used. A similar study supports the idea that comorbidities are associated with higher rates of relapse and mental distress [21]; however, a contradictory result was found by Davis [19], where even though there were high rates of depression, post-traumatic stress disorder, anxiety and bipolar disorders, comorbid diagnoses did not affect the outcomes of detoxification. It is nonetheless important to note that Davis’s study took place in residential treatment settings, whereas the current study took place in the acute setting. Our study found significant differences in symptom profiles across different drug types among individuals who shared the diagnosis of cannabis dependence. Specific pairwise analysis was not possible due to the small cohort size.

The most used substances after cannabis were NPS and unknown substances. A limitation of our study was the partial lack of information about the consumed drugs—several participants did not know what they consumed, and no qualitative tests were conducted, while others did know but it was not adequately registered. Therefore, the actual frequency of NPS could be somewhat higher. Nonetheless, the rate of NPS users was as high as expected, as the EMCDDA’s statistics position Romania in first place in Europe. A key cause of the appearance and popularity of these substances is the prohibitory laws around drug consumption in the post-communist era, which prompted the consumer search for alternatives to cannabis and opioids [22].

Our initial hypothesis was that NPS users have a worse prognosis than non-NPS users, as well as a higher rate of psychotic symptoms. However, this was not confirmed by our findings, as the first group showed no considerable differences in symptom intensity compared to non-users. Furthermore, no specific drug influenced symptom intensity or readmission risk. A significant interaction between drug type and comorbidity was found to influence symptoms, but not readmission. Based on their similar result, Marquez-Arrico et al. concluded that in dual depression, the depressive symptoms play a greater role than SUD in physical functioning and health changes, even though the quality of life of patients with DD is lower than for patients with SUD only [23]. Our findings also partially confirm those of Adan et al., who found greater psychopathological severity, greater attendance at emergency services and a higher frequency of psychiatric admissions by patients with DD [24].

Readmission risk was not found to be significantly increased for the DD group compared to the SUD-only group, which contradicts the results of another study [25].

Disorders related to sedative, hypnotic and opioid use were not common. Strikingly, benzodiazepines were, with two exceptions, only used as a drug of choice. The low rate of opioid use disorders could be explained by a regional trend, in contrast with Bucharest, where 3.3% of acute intoxication cases involved heroin in 2022 (heroin and other opioids—the current situation in Europe) [26]. Unfortunately, data about opioid consumption in the western part of Romania are not available.

The strengths of this study include its uniqueness in the national context, the thorough statistical methods applied and an open attitude to the heterogeneity which defines the very field of psychiatry.

Naturally, this study has limitations as well. Firstly, the reduced cohort size limits the generalization of our results. Secondly, our data were relatively limited in comparison to other studies about addiction due to lack of funding, and this was especially important when discussing drug types. NPS were frequently used but unfortunately little described due to a lack of toxicological tests. Finally, regarding the use of the predictive model, the reported 95% accuracy should be interpreted with caution, as it reflects in-sample performance without cross-validation, which raises the risk of overfitting. While the random forest model was useful for exploring potential predictors, simpler and more interpretable models may also provide valuable insights.

## 5. Conclusions

Our study underlines the increased severity of psychopathology associated with dual diagnosis. Despite the size of the cohort, this study identified three clusters of symptoms which showed even surprising associations and differences between genders. These clinical features were not necessarily classical, which underlines the heterogeneity of psychiatric disorders.

## Figures and Tables

**Figure 1 healthcare-13-02543-f001:**
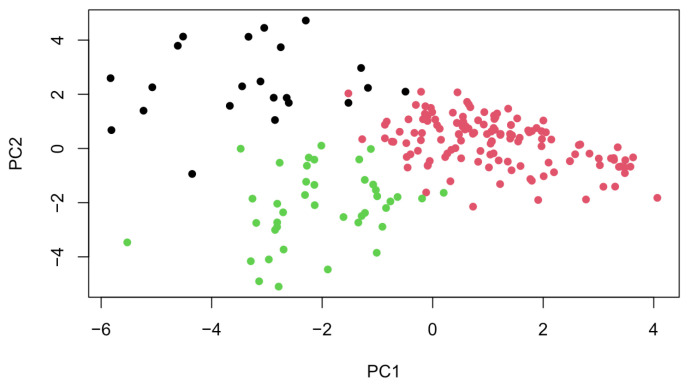
Visualization of PCA results showing three distinct symptom clusters.

**Figure 2 healthcare-13-02543-f002:**
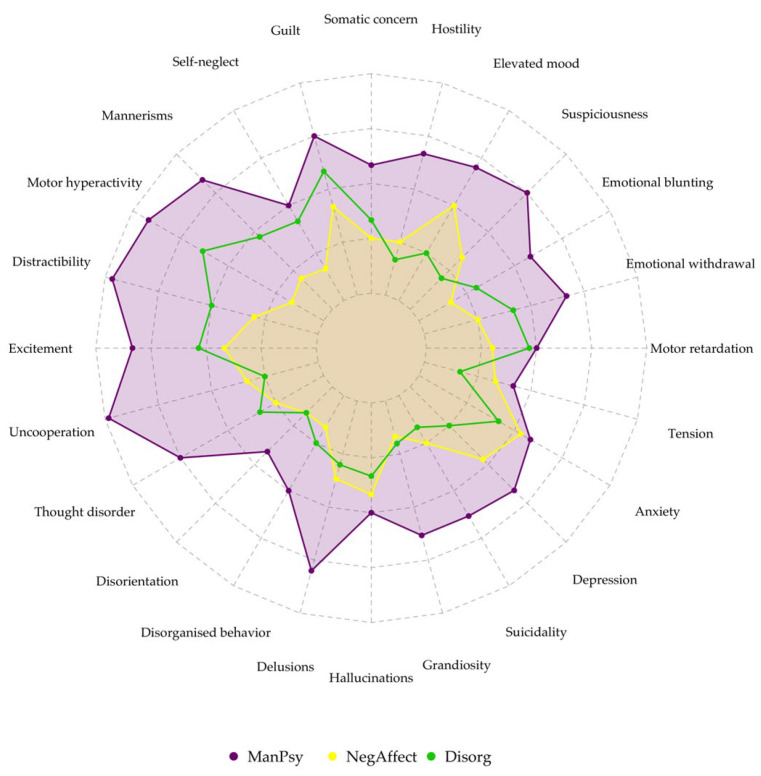
Radar plot showing symptom severity among clusters.

**Figure 3 healthcare-13-02543-f003:**
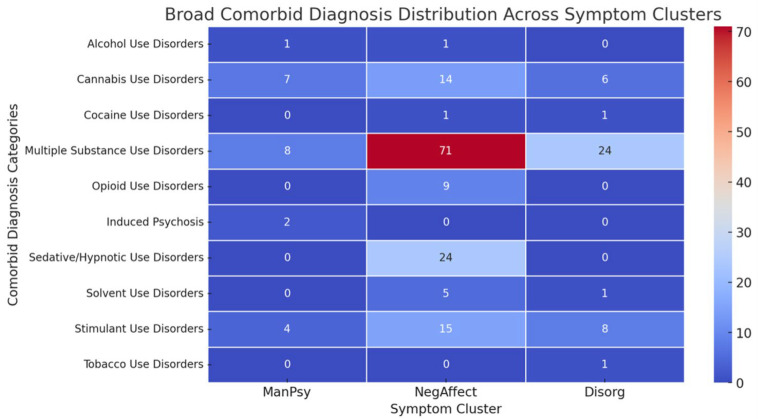
Heatmap of distribution of ICD-10 diagnosis across symptom clusters.

**Table 1 healthcare-13-02543-t001:** Demographic characteristics.

Parameter	Frequency (%, Number)
Gender	
Male	80.8 (164)
Female	19.2 (39)
Occupational status	
Unemployed	40.4 (82)
Legally employed	32.5 (66)
Student	10.3 (21)
Retired due to sickness or age	9.4 (19)
Illegally employed	7.4 (15)
Residence	
Urban area	69 (140)
Rural area	31 (63)
Relationship status	
Single	82.7 (168)
In a relationship	8.9 (18)
Married	7.4 (15)
Divorced	1 (2)

**Table 2 healthcare-13-02543-t002:** Symptoms distributed by gender.

Parameter	Frequency (%, Number)	Distribution by Gender (Male, Female)		X^2^ (*p*-Value)
Positive history of psychiatric illness	72.4 (147)	120	27	0.245 (0.621)
Positive history of depressive episodes	14.8 (30)	16	14	17.096 (<0.001)
Anxiety	36 (73)	52	21	6.706 (0.01)
Irritability	88.7 (180)	147	33	0.79 (0.374)
Emotional lability	54.2 (110)	78	32	15.098 (<0.001)
Sleep disturbances	81.8 (166)	132	34	0.947 (0.331)
Disorders in behavior	47.8 (97)	82	15	1.681 (0.195)
Impulsivity	63.5 (129)	107	22	1.061 (0.303)
Psychomotor agitation	87.7 (178)	145	33	0.421 (0.516)
Self-harm	10.3 (21)	15	6	1.322 (0.25)
Attempted suicides	4.9 (10)	7	3	0.789 (0.374)
Suicidal ideation	11.3 (23)	18	5	0.107 (0.744)
Aggression (all) ^1^	35.9 (73)	69	4	12.5 (0.0004)
Verbal aggression	35.5 (72)	68	4	13.406 (<0.001) ^a^
Physical aggression	18.7 (38)	37	1	0.198 (0.004) ^b^
Induced psychosis	27.1 (55)	45	10	0.052 (0.82)
Delusions (all)	40.9 (83)	70	13	1.14 (0.286)
Persecution delusions	25.6 (52)	47	5	4.148 (0.042)
Interpretation delusions	19.7 (40)	34	6	0.569 (0.451)
Following delusions	16.3 (33)	29	4	0.007 (0.237) ^a^
Spiritual delusions	8.4 (17)	14	3	0.00 (0.863)
Grandiose delusions	9.4 (19)	17	2	0.007 (0.282) ^a^
Depressive delusions	6.4 (13)	6	7	10.735 (0.001)
Poisoning delusions	2.5 (5)	5	0	0.077 (0.27)
Jealousy delusions	1 (2)	2	0	0.049 (0.488)
Hallucinations ^1^	24.6 (50)	41	9	0.002 (0.965)
Auditory	20.7 (42)	35	7	0.221 (0.638)
Visual	12.3 (25)	19	6	0.421 (0.516)
Touch	0.5 (1)	1	0	NA ^2^
Formal thought disorders	59.6 (121)	98	23	0.008 (0.929)
Circumstantiality	14.3 (29)	24	5	0.085 (0.771)
Tangentiality	13.3 (27)	24	3	0.008 (0.225)
Incoherence	6.4 (13)	11	2	0.001 (0.71)
Depressive episode	24.6 (50)	32	18	12.047 (<0.001)
Light depressive episode	7.4 (15)	8	7	15.08 (<0.0001)
Moderate depressive episode	8.9 (18)	13	5	2.287 (0.13)
Severe depressive episode	1 (2)	1	1	NA ^2^
Manic symptoms	6.9 (14)	12	2	0.002 (0.616)
Mixed affective symptoms	2.5 (5)	2	3	0.034 (0.041)
Uncomplicated withdrawal	20.68 (42)	24	18	19.076 (<0.001)
Complicated withdrawal	2.4 (5)	4	1	0.003 (0.964) ^b^

^1^ Actual frequency of individuals with at least one subtype of symptom present, ^2^ Not applicable, ^a^ Yates correction applied, ^b^ Fisher correction applied.

**Table 3 healthcare-13-02543-t003:** Main and secondary consumption of drugs.

Substances	Primary Drug (%, Number)	Secondary Drug (%, Number)
Cannabis	26.1 (53)	5.4 (11)
Unspecified	16.3 (33)	NA ^1^
Methamphetamine	13.8 (28)	2 (2)
Benzodiazepines	12.3 (25)	NA
NPS	11.8 (24)	2 (2)
Amphetamines	6.9 (14)	2 (2)
Opioids	5.4 (11)	2 (2)
Solvents	3.9 (8)	NA ^1^
Cocaine	NA ^1^	3 (6)
Ketamine	NA^1^	2 (2)

^1^ Not applicable.

**Table 4 healthcare-13-02543-t004:** Primary ICD-10 diagnoses and their distribution by gender.

Primary Psychiatric Diagnosis	Frequency (%, Number)	Distribution by Gender (Male, Female)		*p*-Value
Dependence syndrome due to multiple drug use	25.1 (51)	45	6	0.118
Harmful use of multiple psychoactive substances	14.8 (30)	24	6	0.014
Psychosis induced by multiple psychoactive substances	8.9 (18)	17	1	0.13
Dependence syndrome due to sedatives or hypnotics	6.4 (13)	7	6	0.01
Dependence syndrome due to marijuana	5.9 (12)	10	2	1
Harmful use of stimulants	5.4 (11)	10	1	0.69
Dependence syndrome due to stimulants	5.4 (11)	9	2	1

**Table 5 healthcare-13-02543-t005:** Random forest model results. No cross-validation has been performed.

	Prediction of Symptom Clusters	Prediction of Readmission Within a Year	Prediction of Readmission Within the Month
Model accuracy	68.3%	90.2%	95.1%
Feature importance of comorbid diagnosis	60.7%	49.1%	49.2%
Feature importance of primary drug	39.3%	50.9%	50.8%

**Table 6 healthcare-13-02543-t006:** ANOVA results and interpretation.

Analysis	F-Statistic	*p*-Value	Interpretation
Effect of drug type on symptom cluster	0.566	0.898	Drug type alone does not significantly influence the clinical picture.
Effect of drug type on readmission within the year	0.807	0.669	Drug type alone does not significantly predict long-term readmission.
Effect of drug type on readmission within the month	0.945	0.516	Drug type alone does not significantly predict short-term admission.
Effect of comorbidity on clusters	−1.89 × 10^−9^	1.0	Comorbidity alone does not significantly predict cluster membership.
Effect of comorbidity on readmission within the year	1.7 × 10^−10^	1.0	Comorbidity alone does not significantly predict long-term readmission.
Effect of comorbidity on readmission within the month	9.06 × 10^−10^	1.0	Comorbidity alone does not significantly predict short-term readmission.
Interaction effect of drug type and comorbidity on cluster	2.71	<0.0001	Comorbidity significantly interacts with drug type to influence symptoms.
Interaction effect of drug type and comorbidity on readmission within the year	0.716	0.928	No significant interaction between comorbidity and drug type influencing readmission in the long term.
Interaction effect of drug type and comorbidity on readmission within the month	0.702	0.94	No significant interaction between comorbidity and drug type influencing readmission in the short term.

## Data Availability

The data supporting the conclusions of this article will be made available by the authors on request due to internal regulations.

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
