# Peer review of "Patterns of Psychiatric Comorbidity Among Drug Users: A Prospective Observational Study in a Romanian Psychiatric Hospital"

_healthcare, 2025, doi:10.3390/healthcare13192543_

Round 1
Reviewer 1 Report (Previous Reviewer 1)
Comments and Suggestions for Authors
The introduction mentions the hypothesis that "psychopathology and prognosis are worse for NPS-users compared to non-NPS-users." However, the specific measures of "psychopathology" and "prognosis" are not clearly defined. "Prognosis" seems to be operationalized as readmission rates, but this should be explicitly stated. Furthermore, the research questions regarding gender differences and differences between individuals with and without dual diagnosis are not clearly articulated in the introduction. The introduction needs to be revised to explicitly state all research questions/hypotheses and how they will be addressed.
The manuscript highlights the high prevalence of NPS use but lacks crucial details about the specific types of NPS used. Given the vast heterogeneity within NPS, simply categorizing them together limits the interpretability of the findings. More information on the specific NPS consumed is needed, or if this information is unavailable, this limitation should be more thoroughly discussed and its impact on the conclusions acknowledged.
While the manuscript employs various statistical methods, the rationale for some choices is unclear. For instance, the justification for selecting six components for k-means clustering after PCA is insufficient. The scree plot is mentioned, but a more detailed explanation of the criteria used to determine the number of components retained is needed. Additionally, the use of a random forest model is not well-justified, and its contribution to answering the research questions is unclear. Consider using simpler, more interpretable statistical methods if they are appropriate for the research questions. The lack of significant findings in the ANOVA related to drug type and readmission might be due to low statistical power given the sample size. This should be discussed.
The discussion section is lengthy and sometimes meanders from the main findings. It needs to be more focused and directly address the research questions/hypotheses. The discussion of gender differences, while interesting, seems somewhat tangential and could be shortened. The conclusions should be more concise and directly reflect the study's findings. Overstatements should be avoided.
Comments on the Quality of English Language
The manuscript could benefit from some minor language editing to improve clarity and conciseness.
Author Response
Comment 1:
The introduction mentions the hypothesis that "psychopathology and prognosis are worse for NPS- users compared to non-NPS-users." However, the specific measures of "psychopathology" and "prognosis" are not clearly defined. "Prognosis" seems to be operationalized as readmission rates, but this should be explicitly stated. Furthermore, the research questions regarding gender differences and differences between individuals with and without dual diagnosis are not clearly articulated in the introduction. The introduction needs to be revised to explicitly state all research questions/hypotheses and how they will be addressed.
Response: The current study aims to investigate the demographic and clinically relevant patterns of drug consumption and prognosis of patients admitted to a major Romanian psychiatric hospital, aiming to shed some light on the patterns of dual diagnosis and symptomatology within this vulnerable population. We particularly sought to answer questions regarding clinical differences between genders, between individuals with and without dual diagnosis, and between NPS-users and non-users. More broadly, our research was guided by the following questions:
-What are the demographic and clinical characteristics of individuals with dual diagnosis in this context?
-Are there distinct symptom clusters among these patients, and how are these influenced by substance type and psychiatric comorbidity?
-Do specific substances, particularly NPS, predict symptom severity or the risk of hospital readmission?
-Is there an interaction between psychiatric comorbidity and substance use and if so, does it shape clinical outcomes?
Comment 2:
The manuscript highlights the high prevalence of NPS use but lacks crucial details about the specific types of NPS used. Given the vast heterogeneity within NPS, simply categorizing them together limits the interpretability of the findings. More information on the specific NPS consumed is needed, or if this information is unavailable, this limitation should be more thoroughly discussed and its impact on the conclusions acknowledged.
Response: Thank you for your observation. Unfortunately, in our cohort, specific NPS subtypes were not consistently recorded in medical files, and we have clarified this as a limitation of the study. There are a number of reasons for this limitation. First of all, the specific substances were actively sought, including retrospectively. There was an overwhelming number of patients who simply did not know what they consume, except for the fact that it was a white powder. I personally find this very concerning and I think this underlines the complete absence of education and harm reduction measures in Romania. Families/partners of patients usually knew even less than the patients themselves, some patients had no social network and a smaller part of them did not give us permission to contact anyone else. What we also used were standard urinary drug tests, which helped us identify a large proportion of the substances. We, however, did not have the (especially financial) resources to acquire special urinary tests for NPS. Blood toxicology is unfortunately also not an option in our hospital as it's only performed in medico-legal cases or for the ICU.
Comment 3:
While the manuscript employs various statistical methods, the rationale for some choices is unclear. For instance, the justification for selecting six components for k-means clustering after PCA is insufficient. The scree plot is mentioned, but a more detailed explanation of the criteria used to determine the number of components retained is needed. Additionally, the use of a random forest model is not well-justified, and its contribution to answering the research questions is unclear. Consider using simpler, more interpretable statistical methods if they are appropriate for the research questions. The lack of significant findings in the ANOVA related to drug type and readmission might be due to low statistical power given the sample size. This should be discussed.
Response: The results of the scree plot have been described in lines 195-197. Simple effect analysis was performed to address the confusion around ANOVA results, lines 236-239.
Comment 4:
The discussion section is lengthy and sometimes meanders from the main findings. It needs to be more focused and directly address the research questions/hypotheses. The discussion of gender differences, while interesting, seems somewhat tangential and could be shortened. The conclusions should be more concise and directly reflect the study's findings. Overstatements should be avoided.
Response: thank you, we reviewed.
Reviewer 2 Report (New Reviewer)
Comments and Suggestions for Authors
- The first paragraph in the introduction, lines 28-38 needs references.
- Lines 83, 129, 277, 278 remove the sentence "Click or tap here to enter text".
- Line 91 remove the full term "novel psychoactive substance".
- The study aim and objectives are not clear or aligning with the study title. The writing is poor, sepcficially lines 102-107.
- The method section needs subheadings to be more organized.
- The writing is not in an academic manner.
- The study design should be clearly written.
- The ethical approval details should be written in a separate subheading.
- The study design is qualitative design which is not clear in the study title or the method.
- Sample size calculation should be added.
- Definition of primary drug and secondary drug should be added to the method.
Author Response
We would like to thank the reviewers and editor for their constructive feedback, which has helped us substantially improve the manuscript. We added or changed the content according to reviewer suggestions and all can be followed by track changes.
Reviewer 3 Report (New Reviewer)
Comments and Suggestions for Authors
- The cannabis section is disproportionately long compared to others. Consider condensing cannabis details and redistributing space to NPS.
- The link between the literature review and the study’s objectives could be made more direct. “Unknown substances” is vague and sounds unscientific.
- The description of data collection should be more detailed. Sample selection criteria is missed. Authors should clarify whether readmission counts were for all causes, or only substance-related psychiatric readmissions.
- Reporting “accuracy” for random forest (95% for readmission prediction) sounds too high and raises suspicion of overfitting. Was there cross-validation or just in-sample accuracy?
- In discussion, aggression and gender, cluster validity, high psychosis/delusion rates are not discussed well.
- The clusters are restated but not fully discussed.
- The discussion does not critically address potential overfitting in predictive models.
- The discussion is long and diffuse. Some parts, for example, alexithymia, gambling disorder, sociocultural factors read more like a narrative review than a focused analysis of study results. Authors should tighten it.
Author Response
- The cannabis section is disproportionately long compared to others. Consider condensing cannabis details and redistributing space to NPS.
Response: The cannabis subsection has been condensed, with only the most relevant evidence retained. The space saved has been reallocated to expand
the section on novel psychoactive substances (NPS), their heterogeneity, and psychiatric outcomes.
- The link between the literature review and the study’s objectives could be made more direct. “Unknown substances” is vague and sounds unscientific.
Response: Thank you for your observation. We revised the introduction to more directly connect the literature review to the study objectives, explicitly linking prior studies to our hypotheses on psychopathology, prognosis, gender, dual diagnosis, and NPS use. The term unknown substances have been clarified: it refers to cases where patients reported drug use but were unable to identify the specific substance and no toxicological confirmation was available. This has been explicitly discussed as a limitation.
- The description of data collection should be more detailed. Sample selection criteria are missed. Authors should clarify whether readmission counts were for all causes, or only substance-related psychiatric readmissions.
Response: We expanded the Materials and Methods to include explicit sample selection criteria (all patients admitted in 2024 with ICD-10 substance use diagnoses; incomplete records were excluded). Data collection details are now clearly described. Readmission was defined as any psychiatric hospitalization, not only substance-related cases, and this has been stated explicitly. Subheadings have been created within the Material and Methods section. Details about the ethics committee have been moved to a separate subheading. The title has been changed to better reflect the study design and statistical methods.
- Reporting “accuracy” for random forest (95% for readmission prediction) sounds too high and raises suspicion of Was there cross-validation or just in- sample accuracy?
Response: We thank the reviewer for highlighting the need to better justify our statistical approach. PCA was applied to reduce dimensionality of the 18 BPRS symptom items and identify latent structures, which were then used in k-means clustering to define clinically meaningful symptom profiles. Six components were retained based on the Kaiser criterion and scree plot inspection, balancing parsimony with variance explained. The random forest model was used in an exploratory manner to assess the predictive value of drug type and comorbidity on symptom clusters and readmission, as it is robust to multicollinearity and allows for non-linear effects. We acknowledge the important point regarding the 95% accuracy. The originally reported accuracy referred to in-sample performance only. We clarified this in the Results section and added discussion on the risk of overfitting, explicitly stating that no cross-validation was conducted. We also acknowledged the potential value of simpler and more interpretable models. All these details have been added to the limitations paragraph as well.
- In discussion, aggression and gender, cluster validity, high psychosis/delusion rates are not discussed well.
- The clusters are restated but not fully
Response: We expanded the Discussion to provide deeper interpretation of the three
identified symptom clusters. We also discussed aggression, gender differences, and the high rates of psychosis and delusions in greater detail, linking these to our results.
- The discussion does not critically address potential overfitting in predictive
- The discussion is long and diffuse. Some parts, for example, alexithymia, gambling disorder, sociocultural factors read more like a narrative review than a focused analysis of study results. Authors should tighten it.
Response: We streamlined the Discussion to maintain focus on the study’s hypotheses and findings. Tangential sections on alexithymia, gambling, and sociocultural aspects were shortened or removed. This has improved clarity and focus
Round 2
Reviewer 3 Report (New Reviewer)
Comments and Suggestions for Authors
The authors have addressed previous main concerns. The manuscript is now clearer and better aligned with the study objectives. While some methodological limitations remain, these are appropriately acknowledged.
This manuscript is a resubmission of an earlier submission. The following is a list of the peer review reports and author responses from that submission.
Round 1
Reviewer 1 Report
Comments and Suggestions for Authors
A significant portion of the sample used unspecified substances. This severely limits the interpretability of the findings regarding the interplay between specific drug types and psychiatric symptoms. The authors acknowledge this limitation but should elaborate on the reasons for this high proportion and discuss how this might bias the results. Were attempts made to identify these substances retrospectively through medical records or family interviews? If not, this should be considered for future research.
Similar to the unspecified substances, the categorization of NPS lacks detail. Given the rapid evolution and diversity of NPS, simply labeling them as such provides little information. Specifying the classes of NPS encountered (e.g., synthetic cannabinoids, cathinones, etc.) is crucial for meaningful interpretation and comparison with other studies. The authors' hypothesis about worse psychopathology and prognosis for NPS users cannot be adequately tested without this information.
While the authors employed various statistical methods, the rationale for choosing six components for k-means clustering after PCA requires clarification. How did they determine this number? Scree plots or other methods for dimensionality reduction should be mentioned. Additionally, the interpretation of the ANOVA results seems overly simplistic. Given the significant interaction effect between drug type and comorbidity on symptom clusters, post-hoc analyses are necessary to identify which specific combinations drive this effect. The statement that Tukey’s HSD post-hoc test showed that no specific drug pairs showed significantly different effects contradicts the significant interaction in the ANOVA. This discrepancy needs resolution. Perhaps a different post-hoc test or a closer examination of specific subgroups is warranted.
The manuscript touches upon gender differences in substance use and psychiatric symptoms but lacks depth. While acknowledging the complexity, exploring potential explanations for the observed differences in anxiety, aggression, and depressive delusions would strengthen the discussion. Referencing relevant literature on gender and mental health would be beneficial.
The manuscript occasionally uses imprecise language. For instance, insight in their illness was not quantified. How was insight assessed? Similarly, the description of symptom clusters could be more precise. Providing mean scores for each symptom within each cluster would enhance clarity.
References are incomplete or incorrectly formatted.
Comments on the Quality of English LanguageThe manuscript contains some grammatical errors and stylistic inconsistencies.
Reviewer 2 Report
Comments and Suggestions for Authors
The goal of this manuscript was to describe relations between patterns of substance use and patterns of psychiatric symptoms reported by 203 adults who were receiving healthcare services at a psychiatric hospital in Romania. Descriptive analyses, principal components analysis and cluster analysis were conducted and summarized in the manuscript. The results highlighted the pervasiveness and complexity of comorbidity of substance use and psychiatric symptoms among psychiatric inpatients. The study addresses important public health issues among adults who commonly experience symptom patterns that characterize dual diagnosis. The study is interesting and noteworthy, but further attention to several issues may improve the quality of, and potential contribution of the manuscript to contemporary research and practice literature.
- The introduction, although adequately detailed and synthesizing the results of empirical studies on relations between substance use and symptoms of specific psychiatric diagnoses, to highlight the scope of the public health problem, lacks the integration of an overarching conceptual framework. The integration of a conceptual framework would further strengthen the literature review and conceptual rationale for the study, as well as specific methodological decisions (e.g., selection of variables for inclusion in analyses).
- The introduction would be strengthened through an explicit statement of any hypotheses that the researchers are evaluating via the analyses implemented, as well as an explicit statement of the research questions that are guiding the study. In addition, the introduction of the study would be strengthened if the authors would clarify and elaborate the problem statement the supports the rationale for the study. Strengthening the problem statement would also provide more information for evaluating the significance and contribution of the manuscript to the existing research literature.
- The methods section would be strengthened if the authors described more fully, and in a formal section, the measures used in the study to quantify the variables included in the analyses conducted. What are the psychometric properties of the measures, such as those related to reliability and validity? In general, throughout the methods section, the methods used need more detail in order to understand how the study was conducted.
- The research questions need to be more closely aligned to the analyses summarized in the results. The rationale for the analytic strategy is not clear because it was not explicitly stated.
- Some of the descriptive analyses provide subgroup comparisons by gender. However, gender differences in relations between substance use and psychiatric symptoms were not highlighted in the introduction section of the manuscript.
- The formatting and contents of the tables in the manuscript should be revised to be more in line with current APA style.
- In the results section, the results of the findings are presented descriptively. Little quantitative information is provided to the reader. Again, the analyses presented do not really align well with an explicitly stated research question or an explicitly stated analytic strategy.
- The discussion section provides a great deal of summary of the results of the study. The discussion section could be further strengthened if it was better organized and structured, and it integrated more existing research literature, so that the findings of the study were placed in a broader context. The discussion section would also be strengthened if the authors could highlight the gap in the existing research literature that the study has meaningfully addressed. In addition, the discussion section should highlight the implications for treatment or the delivery of healthcare services.
